

# The ESGF Virtual Aggregation (CMIP6 v20240125)

Ezequiel Cimadevilla[1], Bryan N. Lawrence[2], and Antonio S. Cofiño[1]

[1]Instituto de Física de Cantabria (IFCA), CSIC-Universidad de Cantabria, Santander, Spain
[2]National Centre for Atmospheric Science, Department of Meteorology, University of Reading, Reading, UK

**Correspondence:** Ezequiel Cimadevilla (ezequiel.cimadevilla@unican.es)

**Abstract.**

The Earth System Grid Federation (ESGF) holds several petabytes of climate data distributed across millions of files held in data centers worldwide. Obtaining and manipulating the scientific information (climate variables) held in these files is non-trivial. The ESGF Virtual Aggregation is one of several solutions to providing an out-of-the-box aggregated and analysis ready view of those variables. Here we discuss the ESGF Virtual Aggregation in the context of the existing infrastructure, and some of those other solutions providing analysis ready data. We describe how it is constructed, how it can be used, and provide some performance evaluation. It will be seen that the ESGF Virtual Aggregation provides a sustainable solution to some of the problems encountered in producing analysis ready data, without the cost of data replication to different formats, albeit at the cost of more data movement within the analysis than some alternatives. If heavily used, it may also require more ESGF data servers than are currently deployed in data node deployments. The need for such data servers should be a component of ongoing discussions about the future of the ESGF and its constituent core services.

## 1 Introduction

The importance of effective and efficient climate data analysis continues to grow as the demand for understanding the climate system intensifies. Traditionally, climate data repositories have been structured as file distribution systems, primarily facilitating file downloads. However, this conventional approach poses challenges for climate data analysts, requiring them to invest substantial time in managing data access, often unrelated to their ongoing research. The Earth System Grid Federation (ESGF) is a global infrastructure and network that consists of internationally distributed research centers that follows this approach (Williams et al., 2016; Cinquini et al., 2012).

Several methodologies are emerging to streamline climate data analysis, harnessing the potential of Analysis Ready Data (ARD, Dwyer et al. 2018), remote data access and new formats for climate data storage (Abernathey et al., 2021). This paper introduces the ESGF Virtual Aggregation (ESGF-VA), an innovative method for climate data analysis leveraging rarely exploited aspects of the ESGF. It is based on the capabilities of virtual aggregations built on top of the architecture of the ESGF and designed to be included in the federation as an external service. Section 2 provides an overview of the current landscape of climate data analysis and infrastructure. Section 3 introduces the notion of ARD and virtual aggregations in the context of climate data. Following this, Section 4 delineates the methodology employed in the ESGF-VA. Section 5 presents





a performance evaluation of the ESGF-VA comparing it to other data access methods. Section 6 ends with a discussion and concluding remarks.

## 2 Background

In the ESGF, research centers collectively serve as a federated data archive, supporting the distribution of global climate model simulations representing past, present, and future climate conditions (Balaji et al., 2018). The ESGF enables modeling groups to upload model output to federation nodes for archiving and community access at any time. To facilitate multi-model analyses, the ESGF ensures standardization of model output in a specified format. It also facilitates the collection, archival, and access of model output through the ESGF data replication centers. As a result, the ESGF has emerged as the primary distributed data archive for climate data, hosting data for international projects such as CMIP6 (Eyring et al., 2016) and CORDEX (Gutowski Jr. et al., 2016). It catalogues and stores tenths of millions of files, with more than 30 petabytes of data, distributed across research institutes worldwide (Fiore et al., 2021), and it serves as the reference archive for Assessment Reports (AR) (Asadnabizadeh, 2023) on Climate Change produced by the Intergovernmental Panel on Climate Change (IPCC, Venturini et al. (2023)).

The significant growth of data poses a scientific scalability challenge for the climate research community (Balaji et al., 2018). Contributions to the increase in data volume include the systematic increase in model resolution and the complexity of experimental protocols and data requests (Juckes et al., 2020). Although the ESGF infrastructure is designed as a file distribution system, scientific research often requires multidimensional data analysis on datasets encompassing multiple variables, spanning the entire time period, multiple model ensembles and different climate model runs. Figure 1 showcases the result of a data analysis task that encompasses several model runs of a particular model spawning 85 years of data. Several ongoing developments in scientific data research try to address the issues of growing data volume and variety and provide new approaches to data analysis.

Climate Analytics-as-a-Service (CAaaS, Schnase et al. (2016)), GeoDataCubes (Nativi et al., 2017; Mahecha et al., 2020), cloud native data repositories (Abernathey et al., 2021) and Web Processing Services (WPS, 2015) are some of the systems that are being used to improve climate data analysis workflows. The data consolidation process in building these new systems may involve data duplication of an enormous volume of data, incurring in large costs of operational and storage requirements. However, the cost of data duplication is assumed to be compensated by a gain in efficiency in information synthesis. In order to overcome these costs, several technologies do allow the creation of virtual datasets, which provide ARD capabilities without the need to duplicate the original data sources. These provide the opportunity for more sustainable approaches to enhancing climate data analysis capabilities.

The ESGF-VA is one such approach, an innovation aimed to advance the sharing and reuse of scientific climate data by building a catalog of logically aggregated datasets, facilitating remote access to the distributed data hosted in the ESGF. It offers data access (remotely) to convenient and adequate views of the data that allow ad hoc complex queries without the need to duplicate data sources. The ESGF-VA serves as a bridge between the current implementation of the ESGF and the development of climate research cloud native data repositories. Figure 2 shows how it fits into the current ecosystem. The ESGF-VA is



**Figure 1.** Mean Near-Surface Air Temperature for different time periods and different model runs. The code needed to obtain this result is minimal, enabled by the capabilities of the data cube. Because all the information is stored in one single ESGF Virtual Aggregation dataset, only one data source is needed to perform the data analysis. The data is fetched directly from ESGF data nodes on a remote data access basis.

implemented as an additional value-added user service on top of ESGF, running in conjunction with other value-added user
services such as the citation and PID handle services (Petrie et al., 2021). Moreover, to fulfil sustainability requirement, a
balanced strategy is adopted to manage operational costs and complexity. This is achieved by leveraging standard protocols,
technologies, and existing software within the ESGF framework.

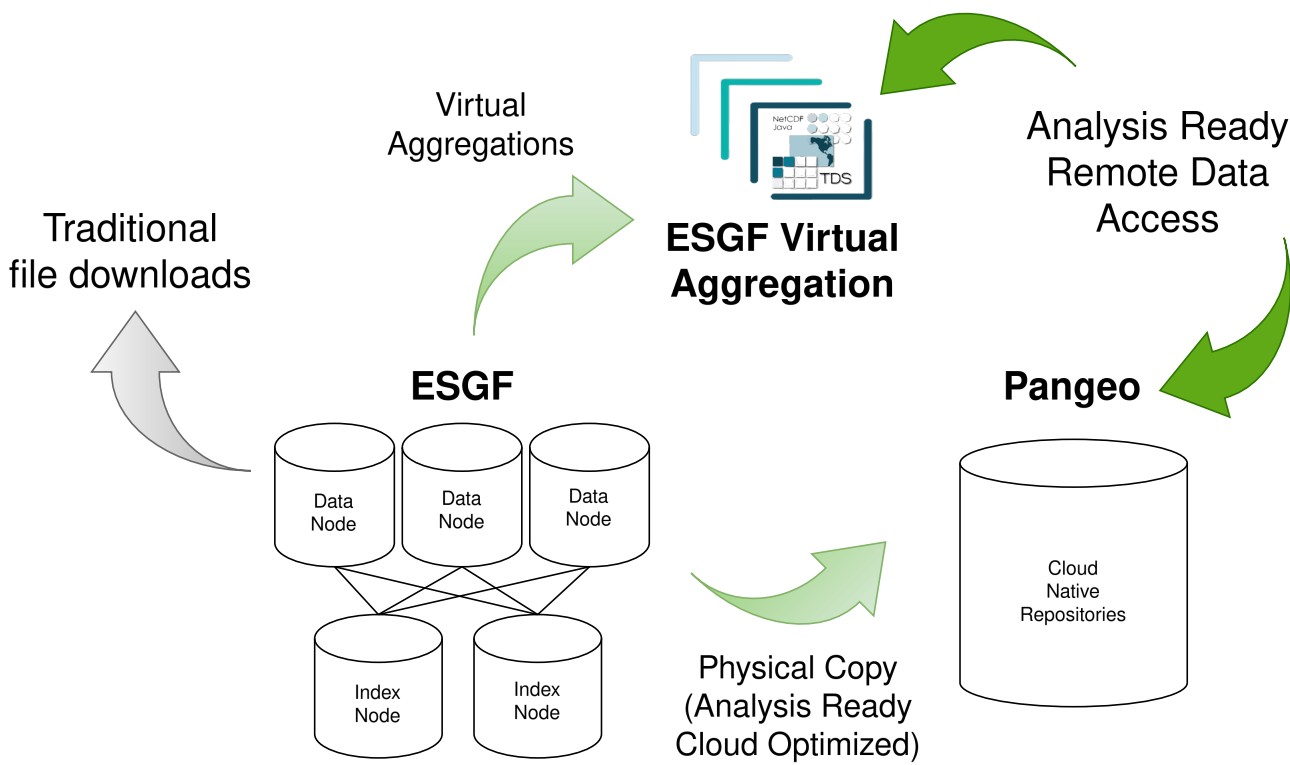

**Figure 2.** The ESGF Virtual Aggregation aims to be a sustainable bridge that eases the technological transition between the current state of the ESGF and more ground-breaking and expensive solutions, based on data replication, such as cloud native repositories.

## 3   Analysis Ready Climate Datasets

ARD refers to datasets that have undergone processing to enable analysis with minimal additional user effort (Dwyer et al.,
2018). The climate data offered by the ESGF is stored in netCDF files (Rew et al., 1989), with an atomic dataset defined as a
set of netCDF files that are aggregated, containing the data from a single climate variable sampled at a single frequency from
a single model running a single experiment (Balaji et al., 2018). These data conform to a file request and a structure controlled
by *Data Reference Syntax* published in partnership with the ESGF. For example, the CMIP6 data conform the CMIP6 data
request, (Juckes et al., 2020), and to the CMIP6 Data Reference Syntax (Taylor et al., 2018).

Figure 3 shows an ESGF atomic dataset and a collection of three netCDF files that conform the dataset. Traditional ESGF-
based climate data analysis workflows involve downloading the files on the collection for at least one atomic dataset. The files
are downloaded to a local workstation or HPC infrastructure. In subsequent steps of the data analysis workflow, developed
software tools and scripts, are executed to perform data analysis tasks. However, these programs must often deal with the





**Figure 3.** ESGF listing three files of a CMIP6 dataset. A common practice in the ESGF consists in splitting the dataset into many files along the time dimension. Smaller files are easier to manage in the federation but performing data analysis becomes harder. This image is a screenshot obtained from the ESGF web portals. Credit is attributed to the ESGF partners supporting these portals. For further details, please refer to https://esgf.llnl.gov/acknowledgments.html.

hierarchical file organization structure of an ESGF repository, introducing complexities unrelated to the primary research analysis task.

The goals of ARD are aimed at addressing the inherent complexities associated with file handling. To achieve this, various methodologies are under consideration, based in either aggregations of the original datasets and/or transition to new infrastructures such as cloud providers. Aggregation-based approaches focus on creating either physical or virtual views of data, optimized for efficient analysis, thereby relieving users from the intricacies of directly manipulating netCDF files. On the other hand, performance optimization-based approaches involve leveraging hardware infrastructures, such as cloud computing providers, to enhance the speed and efficiency of data analysis operations. By utilizing these resources, significant improvements in processing capabilities can be achieved, thereby facilitating smoother data analysis workflows.

ARD based on aggregations can be performed at different layers of abstraction and may involve varying levels of complexity depending on the desired outcome. Many approaches are based on data analysis applications offering functionality for abstracting the underlying files and hierarchical file system organization from the data user. Examples of this approach include xarray's (Hoyer and Hamman, 2017) `open_mfdataset` function and software applications for climate data analysis. Listing 1 provides an example of the usage of the `open_mfdataset` function. Example of software applications include CF-Python (Hassell et al., 2017), xMIP, (Busecke et al., 2023), intake-esm (Banihirwe et al., 2023) and intake-esgf (Collier et al., 2024). In general, these approaches hold an in-memory representation of the virtual dataset or aggregation, which is manipulated by the data analysis package behind the scenes.




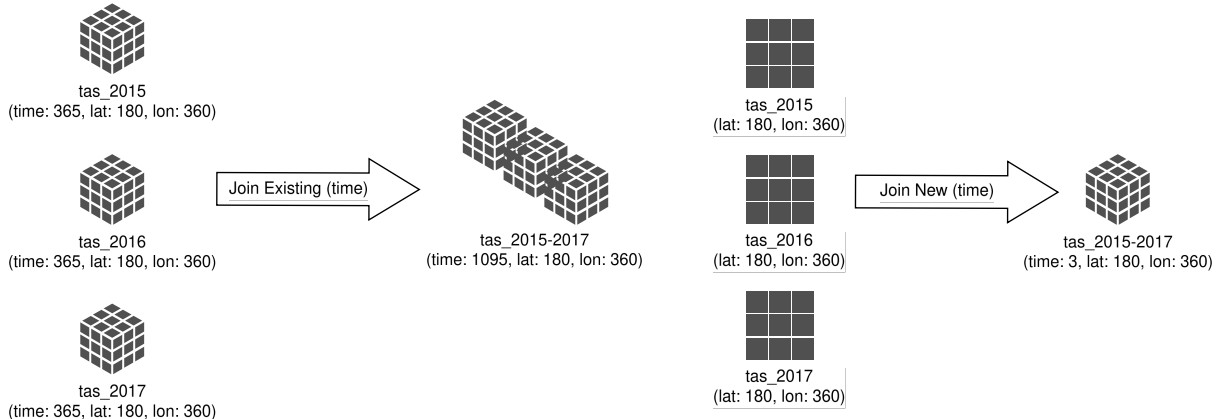

**Figure 4.** Illustration of both *join existing* and *join new* aggregations along the time dimension. In the case of the join exiting, the result of the aggregation is a multidimensional array with the same dimensions (*time*, *lat* and *lon*), in which the size of the time dimension has increased. In the case of the join new aggregation, the time dimension is a new dimension created to aggregate the existing two dimensional arrays into a new three dimensional array.

Software packages may offer to persist their aggregated logical view of the underlying files, but these persistence formats are not interoperable between packages, and/or do not provide interchangeable logical view of aggregation. In the process of generating aggregated views, data may be duplicated, or virtual aggregations can be used to avoid the data duplication. The advantage of relying on virtual dataset capabilities is that data duplication is avoided, and the existing infrastructure may be reused to obtain ARD capabilities without huge costs associated. Examples of virtual aggregations that follow this approach include (but are not limited to) NcML (Caron et al., 2009), Kerchunk (Durant), CFA (Hassell et al.), and HDF5 Virtual Datasets (The HDF Group). The lack of a standard persistence format is also accompanied by different approaches to aggregation methodologies, which arise from a lack of a common data model and a suitable algebra in the context of climate data management.

One attempt to address this issue is the development of the Climate Forecast Aggregation (CFA) conventions, which can describe an aggregated view of netCDF files using the Climate Forecast conventions (Hassell et al.). The CFA conventions provide a formal syntax for storing an aggregation view of file *fragments* using netCDF itself as the storage mechanism. Currently, this syntax is only supported by CF-Python, but libraries and tools are in development to extend CFA support to other packages once the syntax has been through the CF conventions process. CF-Python (Hassell et al., 2017) utilises an underlying data model from the CF conventions, which extends the original netCDF data model with custom structure types. With this data model, a set of unambiguous rules can be established which allow formal manipulation of netCDF variable fragments.





```
1: ds=xr.open_mfdataset(
2:    sorted(glob.glob(
3:      "/storage/ESGF/CMIP6/.../tas_3hr_BCC-CSM2-MR_historical_r1i1p1f1_gn_*.nc")),
4:    combine="nested",
5:    concat_dim=["time"])
```

**Listing 1.** Usage of xarray's *open_mfdataset* to generate an ARD dataset at the application layer from several netCDF files.

Similarly, the software library netCDF-java (Caron et al., 2009) extends the original netCDF data model with additional
operations for manipulation of climate datasets. Using the netCDF-java nomenclature, the operations *join existing* and *join
new* are defined among others. A *join existing* operation concatenates variables of netCDF datasets on a given input dimension. A *join new* operations merges variables of netCDF datasets by creating a new coordinate dimension, thus extending the
dimensionality of the variable. An example of both types of aggregation is shown in Figure 4. Such operations depend on
clean notions of variable identity in order to ensure the semantic correctness of the aggregations. In addition, these virtual
aggregations may be performed by referencing remote sources of data using the OPeNDAP protocol (Garcia et al., 2009). This
particular capability is exploited by the ESGF-VA to provide ARD to the whole ESGF community by exploiting the existence
of OPeNDAP access in the federation (Caron et al., 1997).

```
1: <?xml version="1.0" encoding="UTF-8"?>
2: <netCDF xmlns="http://www.unidata.ucar.edu/namespaces/netCDF/NcML-2.2">
3:   <aggregation dimName="time" type="joinExisting">
4:     <netCDF
5:       location="tas_3hr_BCC-CSM2-MR_historical_r1i1p1f1_gn_195001010000-195212312100.nc"/>
6:     <netCDF
7:       location="tas_3hr_BCC-CSM2-MR_historical_r1i1p1f1_gn_195301010000-195512312100.nc"/>
8:     <netCDF
9:       location="tas_3hr_BCC-CSM2-MR_historical_r1i1p1f1_gn_195601010000-195812312100.nc"/>
10:   </aggregation>
11: </netCDF>
```

**Listing 2.** NcML file that showcases a logical aggregation by performing a *join existing* aggregation over several local netCDF files.

As already discussed, another approach to ARD is to leverage the capabilities of novel hardware infrastructures such as cloud
providers. One notable example of this is the Pangeo initiative, a collaborative effort that brings together diverse communities
to address challenges in climate data analysis. Pangeo has facilitated the development of cloud-native repositories tailored
specifically for climate data analysis needs. These repositories leverage the capabilities of commercial public cloud providers,
such as Amazon Web Services (AWS) or Google Cloud Platform (GCP), to provide scalable, efficient and operational storage
solutions for climate ARD. Pangeo has established a collaboration with the ESGF for further enhancing the accessibility and
usability of climate data for researchers and practitioners worldwide (Abernathey et al., 2021; Stern et al., 2022).





Cloud native repositories have enormously facilitated climate data analysis by leveraging the capabilities of remote data access provided by cloud infrastructures and ARD on top of cloud native data formats. As a result, climate data is accessible

from anywhere and climate data analysts are able to opt for the computation platform of their choice, either HPC infrastructures from their home institutions, user-paid on-demand cloud resources running close to the cloud repository or even a personal laptop. However, the establishment of these repositories has demanded substantial investments in human resources and financial resources to accommodate storage within the premises of cloud service providers[1]. In addition, in order to keep consistent copies of the source repositories, the cost required to sustain cloud native repositories is increased as long as the source

repositories keep updating their datasets. In the following section, the ESGF Virtual Aggregation is described, providing a more sustainable method for providing remote data access and ARD on top of ESGF.

## 4 Implementation

ARD in the form of virtual aggregations or virtual datasets allows users to view the data of their interest as single logical units rather than collections of files. This eliminates the need to navigate through files that necessitate intricate data analysis

programming for interpretation. In ESGF-VA, the logical aggregations are based on aggregation capabilities expressed in NcML, and provided by netCDF-Java. With NcML it's not required to inspect the storage internals of the netCDF files in order to perform the aggregation. This is in contrast to other alternatives such as Kerchunk, which currently requires that all the variables or multidimensional arrays are parameterized with the same configuration (chunking, filters, etc). This is often not the case in ESGF. Kerchunk also needs to extract the byte positions of the chunks from the source netCDF files. Given that the

ESGF contains millions of netCDF files, avoiding inspection of each of them provides an enormous advantage. ESGF index nodes contain metadata about netCDF files, and they can be used to quickly retrieve metadata from the netCDF files. Thus, the complexity and required time of the process of generating the virtual aggregations is reduced in several orders of magnitude.

Remote data access capabilities of the ESGF provided by OPeNDAP and THREDDS (Caron et al., 1997) allow the virtual aggregations to load the data directly and transparently from ESGF data nodes with no file downloads. Figure 5 shows a virtual

dataset of the ESGF Virtual Aggregation (the NcML file) opened with xarray through an OPeNDAP THREDDS data server, since xarray does not currently support opening NcML files directly. Figure 1 shows the result of a data analysis task from this dataset. Because a single dataset contains all the ensemble members of a particular member run, only one dataset is needed to perform this data analysis.

The implementation of the process for generating the virtual aggregations or NcMLs is divided into two steps. First, the

165 search process is responsible for querying and obtaining dataset information and metadata from the ESGF catalog and indexing service and persist it into a local database. Figure 6 gives an overview of the cost in size of generating the NcMLs. Then, the aggregation process queries the local database and creates the virtual datasets for the whole federation. Figure 7 shows how netCDF files from the ESGF that belong to the CMIP6 project are distributed between the virtual datasets. Most of the virtual

---

[1]Refer to Pangeo Showcase talk "How to transform thousands of CMIP6 datasets to zarr with Pangeo Forge - And why we should never do this again!" for further details on this topic.





```
# NcML - 5 netCDF files inside and 1 variable (5 join existing aggregations of 1 file each)
# CMIP6_ScenarioMIP_CNRM-CERFACS_CNRM-CM6-1_ssp245_day_tas_gr_v20190410_aims3.llnl.gov.ncml
ds = xarray.open_dataset(dataset).chunk({"variant_label": 1, "time": 400})
tas = ds["tas"]
tas
```

xarray.DataArray  'tas'  (**variant_label**: 5, **time**: 31411, **lat**: 128, **lon**: 256)

|  | Array | Chunk |
|---|---|---|
| **Bytes** | 19.17 GiB | 50.00 MiB |
| **Shape** | (5, 31411, 128, 256) | (1, 400, 128, 256) |
| **Dask graph** | 395 chunks in 2 graph layers | |
| **Data type** | float32 numpy.ndarray | |

▾ Coordinates:

| lat | (lat) | float64 | -88.93 -87.54 ... 87.54 88.93 | |
|---|---|---|---|---|
| lon | (lon) | float64 | 0.0 1.406 2.812 ... 357.2 358.6 | |
| height | () | float64 | ... | |
| time | (time) | datetime64[ns] | 2015-01-01T12:00:00 ... 2100-12-... | |
| variant_label | (variant_label) | \|S64 | b'r2i1p1f2' ... b'r6i1p1f2' | |

▸ Indexes: (4)

▸ Attributes: (12)

```
%time tas_mean = tas.mean(["lat", "lon"]).compute(num_workers=8)

CPU times: user 642 ms, sys: 28.3 ms, total: 670 ms
Wall time: 11min 18s
```

**Figure 5.** Example of an ESGF Virtual Aggregation NcML file of surface temperatures opened with the xarray package through OPeN-DAP/THREDDS in a Jupyter Notebook. Readers may notice that surface temperature is a three dimensional variable in ESGF, but it is now a four dimensional variable including the model ensemble member dimension. The user does not need to know the number of files involved in the dataset and it can be analysed as a datacube instead of a series of netCDF files. The data requested by the user through xarray will be fetched on demand directly from ESGF data nodes. The NcML is available in appendix A.

datasets contain few references to netCDF files inside (<=100) although some virtual aggregations provide access to hundreds or even thousands of netCDF files. Table 1 shows the ratio of netCDF per NcML for each CMIP6 activity (Eyring et al., 2016).




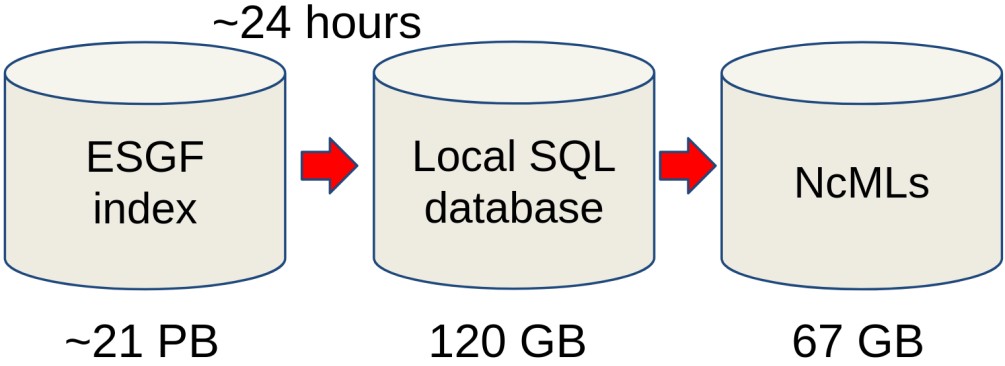

**Figure 6.** From left to right, sizes of the ESGF federation data archive for CMIP6 including replicas, the local database containing the metadata from the federation and the ESGF Virtual Dataset. The ESGF indexes allow querying metadata about netCDF files in a reasonable amount of time. Storage requirement for the virtual aggregations is reduced in several orders of magnitude compared to original data. Other virtual aggregation methods like Kerchunk may expect similar storage requirements.

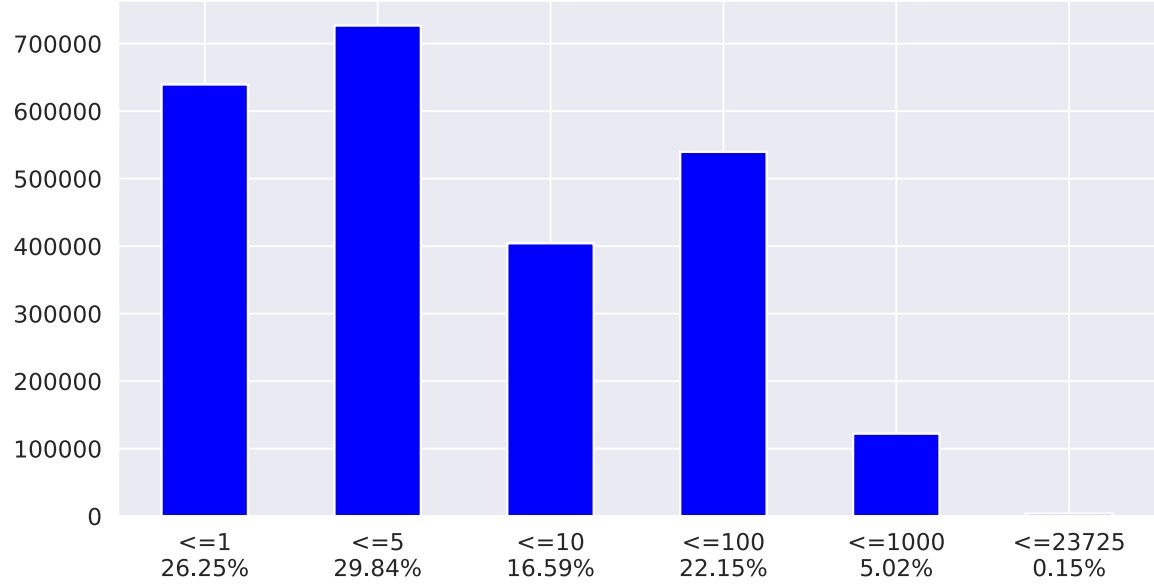

**Figure 7.** Distribution of netCDF files in the virtual datasets (NcMLs). Most of the virtual aggregations are made of a relatively small number of files although some virtual datasets spawn hundreds or thousands of files.





| CMIP6 activity | NcMLs | netCDFs | Ratio (netCDFs / NcML) |
|---|---|---|---|
| ISMIP6 | 2570 | 10864 | 4.23 |
| GMMIP | 9489 | 587501 | 61.91 |
| LS3MIP | 16041 | 188533 | 11.75 |
| OMIP | 17009 | 441578 | 25.96 |
| PAMIP | 19824 | 4931240 | 248.75 |
| CDRMIP | 21189 | 395444 | 18.66 |
| PMIP | 26277 | 645989 | 24.58 |
| GeoMIP | 28470 | 184666 | 6.49 |
| FAFMIP | 41324 | 208881 | 5.05 |
| LUMIP | 57140 | 581573 | 10.18 |
| HighResMIP | 63359 | 5806778 | 91.65 |
| RFMIP | 81548 | 745604 | 9.14 |
| CFMIP | 81599 | 309421 | 3.79 |
| C4MIP | 81964 | 847376 | 10.34 |
| DAMIP | 134708 | 3482721 | 25.85 |
| AerChemMIP | 199307 | 1850392 | 9.28 |
| ScenarioMIP | 250591 | 17317882 | 69.11 |
| CMIP | 505733 | 19090708 | 37.75 |
| DCPP | 506085 | 8152594 | 16.11 |
| **Total** | **2144227** | **65779745** | **-** |

**Table 1.** Number of virtual aggregations (NcMLs), netCDF files for which metadata was retrieved from the federation and ratio of netCDF per NcML generated for CMIP6 in the ESGF Virtual Aggregation. Note that the distribution of number of references to netCDFs files on NcMLs does not follow a uniform distribution (see figure 7).

## 4.1 The ESGF search process

For the search process, the ESGF Search REST API (Cinquini et al., 2012) is used by the client to query the contents of the underlying search index, returning results matching the given constraints in the whole federation. The search service provides useful metadata that allow clients to obtain valuable information about the datasets being queried. However, in the context of

175 the ESGF-VA, it is not as efficient as one would like - sufficient for the first implementation and experiments described here, but in an operational context one would want to see time coordinate information held in the index. This is because applications otherwise need to read such information from each and every file in an aggregation, which may be a significant overhead, before any actual data transfer.

The search process is performed by an iterative querying the ESGF search service, requesting small chunks of data that are

180 manageable by the service. The search service limits the number of records that can be obtained from a single request to ten





thousand elements. Since the federation contains information on the order of tens of millions of records, several requests need to be made. The results are stored in a local SQL database and multiple ESGF Virtual Dataset labels are assigned to the record, in order to identify the virtual dataset in which the records participate in different virtual aggregations.

### 4.2 The aggregation process

The aggregation process is responsible for generating the virtual aggregations and mapping multiple ESGF individual files and their metadata to the appropriate virtual datasets. Although the number of records could be overwhelming, the use of SQL indexes allows the aggregation process to quickly retrieve the granules that belong to the different virtual datasets. The result from the aggregation process in the ESGF Virtual Aggregation is a collection of NcML files that represent the virtual datasets. The virtual datasets are stored in different directories in order to provide appropriate organization. Each virtual dataset is

labeled with the data node to where each of the granules that form the virtual dataset belong. Additionally, the virtual datasets are generated in such a way that replicas from the same virtual dataset are easily identifiable.

The virtual datasets of ESGF-VA are made of two kind of aggregations. First, the ESGF *atomic dataset* aggregation is generated by concatenating the time series of each variable along the time dimension. Figure 4 illustrates this operation and listing 2 provides an NcML example. This concatenation does not increment the rank of dimensions of the multidimensional

array that represents the variable, it only increases the size of the time dimension. This kind of aggregation is ignored in time independent variables such as orography. Then the variables are aggregated by creating a new dimension that represents the variant label (i.e. ensemble members), the different model runs of a climate model. The rank of dimensions is incremented by one, to accommodate a dimension for the ensemble or variant label. It is important to note that for this kind of aggregation to be performed properly, climate variables involved must share a spatial and temporal coordinate reference system, with the

exact same spatial coordinate values. If that were not the case, the resulting multidimensional array would expose incorrect data. Listing 3 shows the NcML for the virtual aggregation in figure 5.

Some issues were found during the development of the ESGF Virtual Aggregation. These involve the usage of the version facet in the publication process of the ESGF and data discrepancies between the ESGF data files and the data cubes offered by the ESGF Virtual Aggregation. In the first place, the version facet is supposed to distinguish between allegedly equal datasets

that have changed due to different kinds of errors, such as incorrect data due to bad model execution or incorrect publication process. In practice, the version facet may, in some cases, end up dividing granules that should belong to the same aggregation, due to inappropriate usage of the facet. From an ESGF-VA point of view this could be avoided by using the latest value of the version facet, but that would lead to issues with maintenance.

### 5 Performance

To investigate the performance of accessing data using the ESGF-VA, an experiment was carried out to examine data access performance from a xarray client. This limited experiment is enough to show some of the benefits of, and issues with, the ESGF-VA. The experiment was carried out with the ESGF-VA utilising OPeNDAP, and for comparison, with Kerchunk aggregation. In

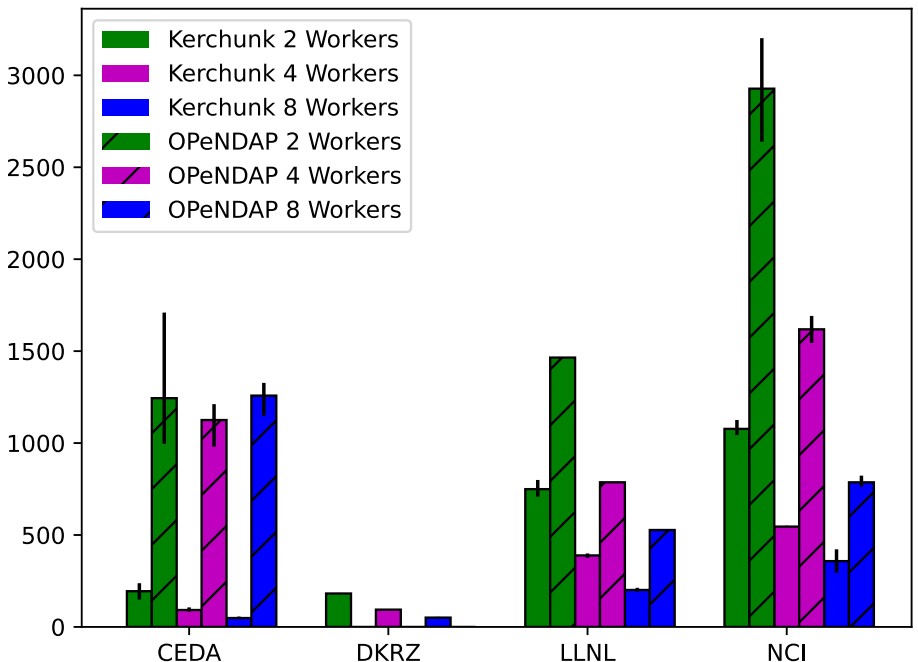

**Figure 8.** The results of the experimental retrieval of data for meaning using Kerchunk and OPeNDAP from a client in Spain (IFCA) to servers in the UK (CEDA), Germany (DKRZ), the US (LLNL) and Australia (NCI). The bars show the mean time taken across experiment replicants for each configuration of number of workers. Where error bars are shown, these reflect the minimum and maximum times taken. Kerchunk data is shown without hatching, and OPeNDAP data with. (Note that there is no OPeNDAP data for DKRZ, and no replicants - and hence no error bars - for the OPeNDAP experiments using the LLNL server.)

both cases, virtual aggregations were generated first, and each was performed with varying numbers of Dask worker processes to test the potential scalability (albeit in a situation where we know that there is limited scalability on the servers themselves, and we believe there would have been little or not contention from other users). Here *Kerchunk* refers to the use of Kerchunk files to access individual blocks of compressed data via Zarr and other Pangeo middleware on the client talking directly to an ESGF HTTPS server, whereas OPeNDAP is the vanilla usage of the ESGF-VA on the client talking to an ESGF OPeNDAP server.

The experiment was simple: we read a dataset consisting of the entire atomic datasets (>80 years) of daily values for one spatially two-dimensional variable (the surface temperature, *tas*) from each member of a simulation and carried out a global mean of that data. The actual calculation was done on a cloud hosted virtual machine in Spain at Instituto de Física de Cantabria (IFCA), while the data was read from each of four ESGF servers. In each case, the dataset was chunked for Dask into segments





of 400 daily values (so each chunk was about 50 MB in memory, the default maximum limit for OPeNDAP), in order to examine the benefit of using multiple Dask workers. We attempted to repeat the experiment five times on each of the ESGF
servers for each of 2,4, and 8 Dask workers. However, it was not possible to get OPeNDAP results from all four servers, or to get a fullset from each of the servers - the reasons for this are discussed below. We did not attempt to mitigate against file system caching in this design, as while it could have impacted on the comparison, in practice the I/O time for reading the data (∼10 GB on disk, ∼20 GB on memory) would be small compared to the overall times reported.

The results are shown in figure 8. There are several obvious results: when using Kerchunk, considerable benefit was gained
by using more workers, and that data nodes close to Spain (where the calculation was done) yielded much faster outcomes than remote data nodes. In each case, OPeNDAP is much slower than Kerchunk, and the benefit of geographical proximity on the OPeNDAP results is much less obvious (e.g. using 8 workers to process data loaded from Australia is faster than using 8 works on data from the UK, but for 2 workers, it is much faster to use the UK data). Unfortunately, DKRZ do not offer the OPeNDAP service, and LLNL took the service down after we did our first experiments and before we added the replicas. It is also clear
that the OPeNDAP results from the CEDA server are anomalous in terms of having no dependency on the number of workers.

As already noted, proximity matters. The benefit of the client-side decompression used by Kerchunk is clear. A priori, we might have expected the OPeNDAP results to be roughly a factor of two slower (given that OPeNDAP decompresses server-side and sends the uncompressed data over the wire), and this is roughly what is seen at LLNL and NCI. As already noted the CEDA OPeNDAP results are anomalous so we make no attempt to explain the disparity between Kerchunk and OPeNDAP
speeds seen there. For this experiment at least, with the fastest times seen (44s and 49s from CEDA and DKRZ respectively), it is clear that the bottleneck is the data flow across the wide area.

Similar experiments with other data highlighted some suboptimal data practices within the ESGF archive. A significant number of CMIP6 datasets stored in the ESGF exhibit poor chunking configurations, specifically related to the time coordinate. Chunking in HDF5 is a crucial technique for optimizing data access performance. It involves organizing how data is stored on
disk, enabling different arrangements based on desired data access patterns. Proper chunking can greatly enhance data access efficiency, similar to how SQL indexes improve database query performance. Conversely, incorrect or inappropriate chunking choices can have a detrimental impact on data access performance. Notably, the CMIP6 files within the ESGF often displayed a chunking configuration of *(1,)* for the time coordinate, resulting in severe degradation of dataset access times (figure 9). Suboptimal chunking configuration negatively affected the efficiency of data retrieval and subsequent analysis tasks. A fix for the
standard climate model output writer (CMOR) has been proposed (https://github.com/PCMDI/cmor/pull/733), although not all modelling centres use CMOR.

## 6 Conclusions

We have introduced the ESGF Virtual Aggregation (ESGF-VA), and shown how it can be used to obtain data from the existing ESGF OPeNDAP servers. In doing so, we have showcased how the ESGF federated index and the ESGF OPeNDAP endpoints
can be used to deliver capabilities beyond conventional file search and download. By enabling remote data analysis over virtual



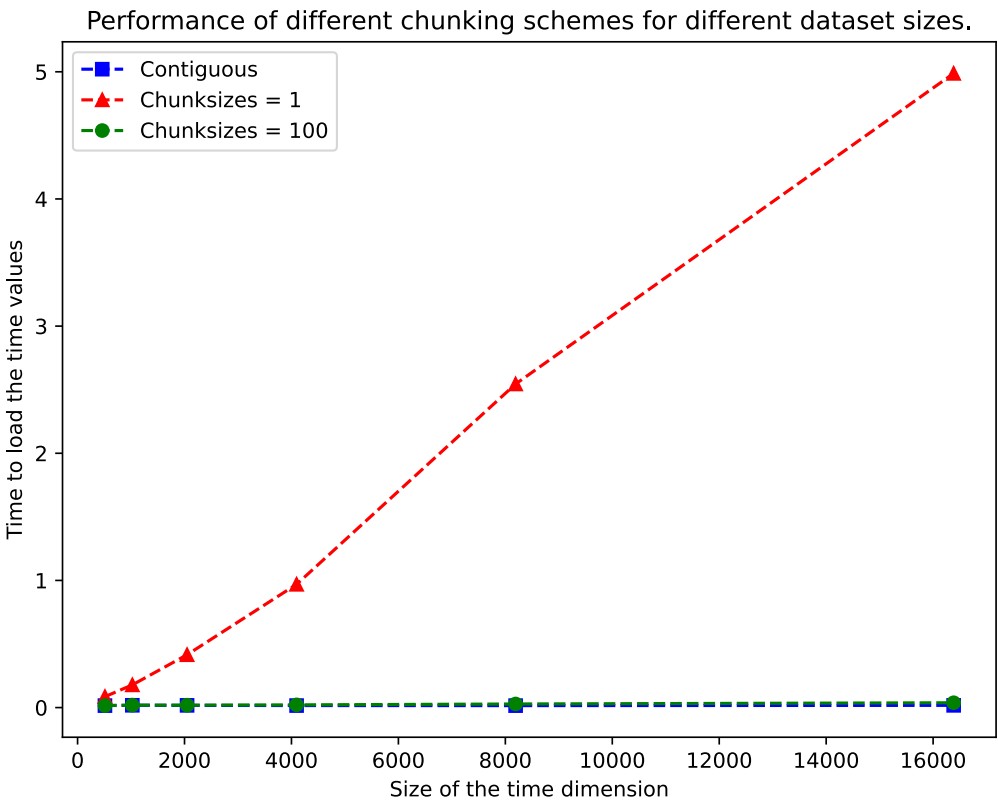

**Figure 9.** Required time to read a temporal coordinate in function of storage type. Contiguous storage does not incur into performance issues. If chunking storage with a bad chunking scheme is used, performance quickly deteriorates.

analysis ready data, the use of the ESGF-VA could enhance the efficiency and productivity of climate data analysis tasks. It could empower researchers to access and analyze data directly within the ESGF environment, eliminating the necessity for time-consuming data transfers and facilitating more streamlined and effective climate data analysis workflows.

While the ESGF-VA provides many benefits for users, albeit with the cost of moving the uncompressed data selections, such benefits would only transpire if there was sufficient server capacity to support demand. Although the ESGF-VA itself requires no change to the ESGF architecture itself, support for access to ESGF data via the OPeNDAP protocol is currently delivered by the use of THREDDS Data Server (a Java web application). While scaling out server infrastructure with THREDDS is possible, it requires both sufficient hardware and significant configuration knowledge. The pros and cons of wider usage of the ESGF-VA or similar OPeNDAP based tools and the consequential need for server capacity and issues of configuration should
form part of future ESGF discussions.



The virtual datasets provided by the ESGF-VA facilitate an aggregated view of the time series as well as the ensemble model members of a particular model. Thus, data analysis comparing different runs of the same model can be performed by loading only the view of one dataset. In doing so, the details of the aggregation are hidden completely from the user, who sees the dataset as a single netCDF. Using the OPeNDAP endpoints of the federation, data analysis can be performed from anywhere.

While this implementation of the ESGF-VA exploits NcML and netCDF-java, the concept is readily extensible across any netCDF client which supports OPeNDAP - i.e. any which utilise the netCDF library itself, rather than directly using HDF5. However, because OPeNDAP performs chunk decompression in the server, it is not as efficient as other data access methods as more data is sent over the network.

The creation of the virtual aggregations presented in this work follows a much more maintainable approach than alternatives

focusing on duplication of the data, such us cloud native repositories. The storage requirements of the virtual aggregations are minimal compared to the relative size of the raw data. In addition, the generation of the virtual aggregations can be performed in few hours, where most of the time is spent querying the ESGF distributed index. As the ESGF-VA aggregation information is obtained directly from the existing ESGF index, it can be generated much faster than the process needed to generate Kerchunk indices, which requires access to each file. The speed of creation of new virtual aggregations, coupled with the lack of actual

data duplication, means that the system can cope well with an environment where datasets are being updated as data processing issues are found and fixed, since the ESGF-VA can be quickly updated. However, whatever system is used to create analysis ready data, it is necessary to know that such updates are necessary - it would be helpful for a future ESGF to have some sort of automated alert system for data updates.

The work on the ESGF-VA has led to the identification of some issues in the ESGF data itself. There is often inconsistent

use of the version facet, and a significant portion of the data stored in the federation does not adhere to best practices regarding the chunking of HDF5. There may be value in both providing better guidance to modelling centres about how to use version facets and in adding some chunk checking to future ingestion processes.

The performance analysis presented in this work suggests a declining interest from the ESGF community in supporting OPeNDAP, given the instability of this service compared to data access based on HTTP. While we do not know the details of

290 the individual server configurations, the fact that the CEDA OPeNDAP results are so odd, and that both DKRZ and LLNL no longer offer OPeNDAP servers, it is plausible to conclude that it is a) difficult to deploy OPeNDAP and b) currently not enough usage to justify it. However, our results suggest that there may yet be mileage in deploying properly configured OPeNDAP services in the future ESGF (maybe with a different server, such as that of Gallagher et al. 2022) - at least until such time that remote direct access to chunks via HTTP is available to a much greater proportion of netCDF clients. In doing so, the

295 use of HTTP compression could mitigate the issue of server side decompression of the chunks. This functionality is currently supported by netCDF clients but is currently provided by few, if any, ESGF nodes. Finally, it would also be helpful if the time coordinate information could be stored in the ESGF index to be used by virtual aggregation clients in a way to avoid the need to read time coordinate values from each file when opening the virtual dataset.

It is clear that the ESGF has and will evolve. Our work suggests the the ongoing evolution of the ESGF needs to address not

only indexing and data download, but where possible, the provision of direct data access suitable for a wide range of use-cases.



Such support may include giving modelling centres good guidance as to how to chunk and organise their data, beyond just relying on CMOR, as not all centres use CMOR.

*Code availability.* https://doi.org/10.5281/zenodo.12179904

## Appendix A: NcML example

**Listing 3.** NcML generated by the ESGF Virtual Aggregation.

```xml
1: <?xml version="1.0" encoding="UTF-8"?>
2: <netcdf xmlns="http://www.unidata.ucar.edu/namespaces/netcdf/ncml-2.2">
3:     <explicit/>
4:     <attribute name="size" type="int" value="11536844671"/>
5:     <attribute name="size_human" value="10.7 GiB"/>
6:
7:     <attribute name="__info__"
8:              value="Virtual dataset generated by the ESGF Virtual Aggregation"/>
9:     <attribute name="__license__"
10:             value="This is a derived dataset product from ESGF, same licenses from original
     datasets apply for this dataset."/>
11:
12:     <!-- only mandatory (when required? = always) attributes from
13:          http://cerfacs.fr/~coquart/data/uploads/cmip6_global_attributes_filenames_cvs_v6.2.6.pdf
14:          are included -->
15:     <!-- mandatory attributes are extracted from netcdf files
16:          BUT creation_date and further_info_url have got custom values relative to EVA -->
17:     <attribute name="activity_id" value="ScenarioMIP"/>
18:     <attribute name="Conventions" value=""/>
19:     <attribute name="data_specs_version" value=""/>
20:     <attribute name="experiment" value=""/>
21:     <attribute name="experiment_id" value="ssp245"/>
22:     <attribute name="forcing_index" value=""/>
23:     <attribute name="frequency" value="day"/>
24:     <attribute name="grid" value=""/>
25:     <attribute name="grid_label" value="gr"/>
26:     <attribute name="initialization_index" value=""/>
27:     <attribute name="institution" value=""/>
28:     <attribute name="institution_id" value="CNRM-CERFACS"/>
29:     <attribute name="license" value=""/>
30:     <attribute name="mip_era" value="CMIP6"/>
31:     <attribute name="nominal_resolution" value=""/>
32:     <attribute name="physics_index" value=""/>
```

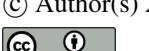



```
33:      <attribute name="product" value="model-output"/>
34:      <attribute name="realization_index" value=""/>
35:      <attribute name="realm" value="atmos"/>
36:      <attribute name="source" value=""/>
37:      <attribute name="source_id" value="CNRM-CM6-1"/>
38:      <attribute name="source_type" value=""/>
39:      <attribute name="sub_experiment" value=""/>
40:      <attribute name="sub_experiment_id" value="none"/>
41:      <attribute name="table_id" value="day"/>
42:      <attribute name="variable_id" value="tas"/>
43:      <attribute name="cmor_version" value=""/>
44:      <!-- attributes that default to "no parent" if they don't exist -->
45:      <attribute name="branch_method" value="no parent"/>
46:      <attribute name="parent_activity_id" value="no parent"/>
47:      <attribute name="parent_experiment_id" value="no parent"/>
48:      <attribute name="parent_mip_era" value="no parent"/>
49:      <attribute name="parent_source_id" value="no parent"/>
50:      <attribute name="parent_time_units" value="no parent"/>
51:      <!-- attributes that are omitted if they don't exist -->
52:
53:      <attribute name="further_info_url" value="See netCDF variable 'further_info_url'"/>
54:      <attribute name="creation_date" value=""/>
55:      <attribute name="version" value="v20190410"/>
56:      <attribute name="replica" value="1"/>
57:
58:      <dimension name="nfiles" length="5"/>
59:      <dimension name="file" length="2"/>
60:      <variable name="further_info_url" type="string" shape="nfiles file">
61:         <values>http://aims3.llnl.gov/thredds/dodsC/css03_data/CMIP6/ScenarioMIP/CNRM-CERFACS/CNRM-
      CM6-1/ssp245/r2i1p1f2/day/tas/gr/v20190410/tas_day_CNRM-CM6-1_ssp245_r2i1p1f2_gr_20150101
      -21001231.nc https://furtherinfo.es-doc.org/CMIP6.CNRM-CERFACS.CNRM-CM6-1.ssp245.none.r2i1p1f2
      http://aims3.llnl.gov/thredds/dodsC/css03_data/CMIP6/ScenarioMIP/CNRM-CERFACS/CNRM-CM6-1/ssp245/
      r3i1p1f2/day/tas/gr/v20190410/tas_day_CNRM-CM6-1_ssp245_r3i1p1f2_gr_20150101-21001231.nc https:
      //furtherinfo.es-doc.org/CMIP6.CNRM-CERFACS.CNRM-CM6-1.ssp245.none.r3i1p1f2 http://aims3.llnl.
      gov/thredds/dodsC/css03_data/CMIP6/ScenarioMIP/CNRM-CERFACS/CNRM-CM6-1/ssp245/r4i1p1f2/day/tas/
      gr/v20190410/tas_day_CNRM-CM6-1_ssp245_r4i1p1f2_gr_20150101-21001231.nc https://furtherinfo.es-
      doc.org/CMIP6.CNRM-CERFACS.CNRM-CM6-1.ssp245.none.r4i1p1f2 http://aims3.llnl.gov/thredds/dodsC/
      css03_data/CMIP6/ScenarioMIP/CNRM-CERFACS/CNRM-CM6-1/ssp245/r5i1p1f2/day/tas/gr/v20190410/
      tas_day_CNRM-CM6-1_ssp245_r5i1p1f2_gr_20150101-21001231.nc https://furtherinfo.es-doc.org/CMIP6.
      CNRM-CERFACS.CNRM-CM6-1.ssp245.none.r5i1p1f2 http://aims3.llnl.gov/thredds/dodsC/css03_data/
      CMIP6/ScenarioMIP/CNRM-CERFACS/CNRM-CM6-1/ssp245/r6i1p1f2/day/tas/gr/v20190410/tas_day_CNRM-CM6
      -1_ssp245_r6i1p1f2_gr_20150101-21001231.nc https://furtherinfo.es-doc.org/CMIP6.CNRM-CERFACS.
      CNRM-CM6-1.ssp245.none.r6i1p1f2</values>
62:      </variable>
```







```
89:            <netcdf coordValue="r5i1p1f2">
90:                <aggregation type="joinExisting" dimName="time">
91:                    <netcdf location="http://aims3.llnl.gov/thredds/dodsC/css03_data/CMIP6/
       ScenarioMIP/CNRM-CERFACS/CNRM-CM6-1/ssp245/r5i1p1f2/day/tas/gr/v20190410/tas_day_CNRM-CM6-1
       _ssp245_r5i1p1f2_gr_20150101-21001231.nc"/>
92:                </aggregation>
93:            </netcdf>
94:            <netcdf coordValue="r6i1p1f2">
95:                <aggregation type="joinExisting" dimName="time">
96:                    <netcdf location="http://aims3.llnl.gov/thredds/dodsC/css03_data/CMIP6/
       ScenarioMIP/CNRM-CERFACS/CNRM-CM6-1/ssp245/r6i1p1f2/day/tas/gr/v20190410/tas_day_CNRM-CM6-1
       _ssp245_r6i1p1f2_gr_20150101-21001231.nc"/>
97:                </aggregation>
98:            </netcdf>
99:        </aggregation>
100: </netcdf>
```

*Author contributions.* **Ezequiel Cimadevilla:** Investigation; methodology; software; visualization; writing – original draft; writing – review and editing.

*Author contributions.* **Bryan N. Lawrence:** Methodology; visualization; writing – original draft; writing – review and editing

*Author contributions.* **Antonio S. Cofiño:** Writing – review and editing

*Competing interests.* The authors declare that they have no competing interests.

*Acknowledgements.* Grant PRE2021-097646 funded by MICIU/AEI/10.13039/501100011033 and by ESF+.



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
