# Peer review of "The ESGF Virtual Aggregation (CMIP6 v20240125)"

_Geoscientific Model Development, 2024_

## Author Response (AR1)

**Response to Anonymous Referee #1**

https://doi.org/10.5194/gmd-2024-120-RC1

> *While the current focus on database operations might make the paper more suitable for a data management journal, there is potential for it to fit within Geoscientific Model Development if the numerical models of the Earth system aspects are emphasized. If the primary contribution remains database-oriented, the manuscript might not meet the scientific inquiry expectations of Geoscientific Model Development. Reframing the study to address earth system model processes more directly could improve its suitability, but if that is not feasible, the authors might consider submitting to a journal more focused on database management and database query. Please refer to the aims and scope of GMD*
> *https://www.geoscientific-model-development.net/about/aims_and_scope.html*

We thank the referee for reviewing the manuscript. To clarify the connection with model evaluation, we have added an entirely new section to the manuscript titled "Model Evaluation". This section presents a use case of model evaluation based on the agreement among model ensemble members. Specifically, it examines the agreement of different model runs of the CanESM5 GCM on precipitation data for two future scenarios relative to the historical reference period. Additionally, we explain the role of virtual datasets and remote data access of the ESGF-VA in implementing this task, highlighting how they eliminate the need for users to download files. We have also added the corresponding notebook to the GitHub repository and created a new release with a persistent identifier, as detailed in the Code Availability section.

**Response to Anonymous Referee #2**

https://doi.org/10.5194/gmd-2024-120-RC2

> *The problem with respect to the dependency of the described solution on available opendap servers (not forseen in the future ESGF infrastructure planing) is described. A short comment on how other types of lightweight data servers e.g. based on xpublish would be an option for the future would be helpfull. Also a short comment on the nature of this dependency would be helpfull - DMR++ Opendap is not mentioned etc. ?*

Thank you for your thorough review of the manuscript. We now mention and include as future work the evaluation of other types of lightweight data servers and metadata files (xpublish and DMR++). We agree with the referee that interest in maintaining OPeNDAP servers within ESGF is decreasing. The purpose of this work has been to provide a justification for their deployment and to offer insights into their usage that may have been overlooked by the community.

> *The manuscript focuses on the distributed database and data management aspects and not so much on the model development, yet there are important implications for the climate model components responsible for generating standardized model (meta)data to enable data distribution e.g. in ESGF (e.g. cmor, versioning, chunking and aggregation problems with the currently available ESGF metadata are mentioned in the paper). With a more explicit description on this aspect it might fit better within Geoscientific Model Development.*

To clarify the connection with model evaluation, we have added an entirely new section titled "Model Evaluation". This section includes a use case that demonstrates model evaluation through agreement among model ensemble members. Specifically, we assess the agreement of different CanESM5 GCM runs on precipitation data for two future scenarios compared to the historical reference period. We have also included the corresponding notebook in the GitHub repository and generated a new release with a persistent identifier, as described in the Code Availability section.

> *The numbers in figure 6 are a bit misleading because numbers refer to different aspects. The illustrated ESGF index is much smaller (GB scale) then the mentioned ~21 PB of data this index is addressing. Yet the ESGF index is used to generate the local sql database. The aspect that other aggregation methods like kerchunk would need to inspect the indexed ~21 PB data is probably something which should not be included in the figure ..*

We have clarified in the image caption that the 21 PB size refers to the raw data of the netCDF files, while the metadata information stored in the index is on the order of gigabytes. This distinction is intended to highlight for readers the several orders of magnitude difference between the raw netCDF files and NcML (or other possible) metadata files. Additionally, we have removed the comment about Kerchunk from the figure caption.

**Response to Astrid Kerkweg**

https://doi.org/10.5194/gmd-2024-120-CEC1

> *However, it would be much appreciated, if you could provide some more information on what is contained in the zenodo directory_ code ,plot scripts, data  etc.*

We thank you for the review of our manuscript. We now include a description of the contents of the repository in the Code Availability section. This section now outlines the Python scripts enabling users to reproduce the ESGF-VA, along with the notebooks used to replicate the results and figures presented in the manuscript. Additionally, we have provided Kerchunk files to facilitate the reproduction of performance analysis results.

> *Furthermore, the DOI should be presented in a form of proper citation.*

We have added proper data to the Zenodo DOI form to make it proper for citation. Also, we have generated a new release with the contents of the repository that includes new contents such as the model evaluation notebook, also described in the Code Availability section.

---

## Author Response (AR2)

**Response to Xiaomeng Huang**

We thank the topic editor for reviewing the manuscript and greatly appreciate the comments. We believe that the review process has resulted in an improved manuscript. Below, we provide responses to all the comments.

1.  *Introduction and Background: The introduction provides a good start, but it could more explicitly state the research gap that ESGF-VA aims to fill. The background section is comprehensive, yet consider reorganizing it to flow more smoothly into the discussion of ESGF-VA. For example, group related information about climate data challenges and existing solutions more coherently.*

We have reorganized the Introduction and Background sections to better highlight the research gap that ESGF-VA aims to address. The revised introduction emphasizes the current challenges faced by the ESGF infrastructure and outlines how we propose to address them. Additionally, the background section has been restructured to ensure a more natural flow within the overall text.

2.  *Methodology and Implementation: The description of ESGF-VA's methodology and implementation is detailed, but some steps could be further broken down for clarity. For instance, in the search and aggregation processes, use sub-sections or bullet points to highlight key operations and how they contribute to the overall functionality.*

We have restructured the Implementation section to provide greater clarity regarding the ESGF-VA's implementation. Bullet points have been added to introduce the two main implementation components of ESGF-VA, preceding the subsections. These bullet points are intended to highlight the scope of both operations in generating the virtual aggregations within ESGF-VA.

3.  *Results and Discussion: When presenting the results, it would be beneficial to have a more in-depth analysis of how they directly support the claims about ESGF-VA's advantages. In the discussion section, better connect the findings back to the broader context of climate data analysis and future research directions. Additionally, consider adding a subsection that summarizes the key takeaways from the study.*

We have reorganized the Conclusions section to better distinguish the research findings from the discussion of various topics related to ESGF. To enhance clarity, we have introduced two subsections: *Summary of Findings* and *Discussion*. The *Summary of Findings* describes how ESGF-VA integrates into the broader ecosystem and emphasizes its specific contributions. The *Discussion* subsection provides insights for the ESGF community on ways to further enhance the federation beyond file search and download, drawing on the experience gained during the development of ESGF-VA.